# Field Evaluation of RD6 Introgression Lines for Yield Performance, Blast, Bacterial Blight Resistance, and Cooking and Eating Qualities

**Myo San Aung Nan, Jirayoo Janto, Arthit Sribunrueang, Tidarat Monkham, Jirawat Sanitchon and Sompong Chankaew \*** 

Department of Agronomy, Faculty of Agriculture, Khon Kaen University, Khon Kaen 40002, Thailand; myo081159@gmail.com (M.S.A.N.); kamong285@gmail.com (J.J.); arthitsribunrueang@gmail.com (A.S.); tidamo@kku.ac.th (T.M.); jirawat@kku.ac.th (J.S.)

**\*** Correspondence: somchan@kku.ac.th; Tel.: +66-8512-40427

**Abstract:** Glutinous rice cultivar "RD6" is well known for its fragrance and high cooking and eating qualities, and is the most popular glutinous cultivar in the north and northeastern regions of Thailand. However, it's susceptible to blast and bacterial blight (BB) diseases. Previously, four blast resistance QTLs on chromosomes 1, 2, 11, and 12, and a single BB resistance gene *xa5* pyramided to the background of the RD6 cultivar were tested for a broad spectrum of disease resistance under greenhouse conditions. In the present study, a field experiment was conducted during the rainy seasons of 2015, 2016, 2017, and 2018, across three locations, for performance evaluations of promising lines in terms of disease reaction, agronomical characteristics, grain yield, and quality attributes. The results revealed that the ILs ($BC_2F_5$ 2-7-5-36, $BC_2F_5$ 2-7-5-43, $BC_2F_5$ 2-8-2-25, and $BC_2F_5$ 6-1/15-2-11) exhibited higher level resistance to leaf blast and neck blast disease. The $BC_2F_5$ 2-8-2-52 showed resistance to both blast and BB diseases and, like all ILs, exhibited superior yield compared to the original RD6. Furthermore, the agronomic traits and grain qualities were similarly displaced, and were therefore recommended as near-isogenic lines to the RD6. This clearly demonstrated that farm phenotypic selection plays an important role in achieving not only NIL resistance to diseases, but also high yield potential, as well as representing an effective way in which to enhance BB, leaf blast, and neck blast resistance in rice planting in the north and northeastern regions of Thailand.

**Keywords:** marker-assisted selection; resistance gene; quantitative trait locus; pseudo-backcrossing

## 1. Introduction

Rice (*Oryza sativa* L.) is a crucial food crop for global food security. Approximately 90% of the world's rice is produced and consumed in Asia [1]. Globally, Thailand ranks as the second largest rice exporter, grown on some 11M ha of land; more than half of the rice grown in Thailand is grown in the north and northeastern regions [2]. Additionally, Thailand has huge varietal diversity and varied rice growing ecologies. Among them, the RD6 improved glutinous rice variety, released in 1977, is one of the most popular lowland varieties for domestic consumption, especially in Thailand's north and northeastern regions [3]. This variety, representing 83% of the northeastern region's total glutinous rice in 1995 [4], is preferred among Thais due to its cooking and eating characteristics, e.g., slender kernel grain, aroma, stickiness, and taste. Currently, it is being cultivated in large areas of lowland, rain-fed terrain in north and northeastern Thailand [3].

However, RD6 rice contains several production constraints, including biotic stresses, among which rice blast, caused by the fungus *Magnaporthe oryzae* [5], and bacterial blight disease, caused by

*Xanthomonas oryzae pv. oryzae* [6], are the most severe, causing high losses in both yield and quality. Rice blast disease can infect all parts of the rice plant including the roots [7]; however, the leaves and neck panicles are the most seriously affected. Leaf blast lesions can reduce the net photosynthetic rate per leaf. Neck blast, considered to be the most destructive phase of the disease, can occur without being preceded by severe leaf blast, resulting in major losses of yield and grain quality [8]. The specific genetic mechanisms of resistance to leaf blast and panicle neck blast may be varied, as well as independently controlled by different *R* genes [9]. Therefore, evaluations of seedling blasts (ignoring panicle blast) have proven inconclusive. Likewise, BB is also a serious disease which affects the seedling to the tillering stage [10]. Previous studies have reported that both diseases occur in more than 80 rice growing countries, resulting in yield losses estimated at more than 50% of production [10,11].

Both blast and BB diseases can be controlled with chemicals, but the spraying of chemicals is hazardous to the environment, causes land degradation, and in turn, increases production costs. While cultural practices, such as the reduction of nitrogen fertilizers and water management, can minimize damage, these practices cannot control the infection completely. Therefore, the utilization of rice varieties carrying resistance genes has proven to be one of the most economical, effective, and environment-friendly strategies for the management of rice diseases [12]. Gene backcross pyramiding combines one or more desirable genes from multiple parents into a single genotype, without compromising on agronomic performance and grain qualities [13]. In earlier studies, Suwannual et al. [14] and Pinta et al. [15] successfully introgressed one BB resistant gene (*xa5*) and four blast resistant QTLs in chromosomes 1, 2, 11, and 12 through marker-assisted selection. Broad spectrum resistance was then tested using artificial inoculum with eight blast isolates and ten BB isolates under greenhouse conditions [15]. However, IL information on the agromorphology traits, disease resistance levels, grain yield, as well as cooking and eating quality, has not yet been reported. Therefore, the performance of a farm-level, field-based trial emphasizing agromorphological traits, disease resistance levels, and grain and cooking quality is essential. The objective of this study was therefore to investigate the performance of introgression lines in paddy fields, and their grain quality attributes following the concept of backcross selection, in order to select superior, improved RD6 NIL lines which are resistant to blast and BB diseases, and which offer high yield potentials and attribute qualities.

## 2. Materials and Methods

### 2.1. Plant Materials and Experimental Design

The ten selected RD6 introgression lines (ILs) [14,15], ILs without QTLs, four check lines including three donor parents (Jao Hom Nin (JHN), P0489, and IR62266), and a recurrent parent (RP) RD6, were evaluated for blast, BB diseases reactions, yield, yield components, and grain quality attributes (Table 1). Field experiments were conducted in 2015, 2016, 2017, and 2018 across three locations (Figure 1). All entries were assigned in a randomized complete block design (RCBD) with three replications during the rainy season of each year and location. Natural infection was applied by sowing the KDML105 as a susceptible variety in the experiment surroundings one month prior to transplantation. Each plot contained five rows, 1.5 m in length, with 25 cm × 25 cm between and within each row. To ensure a uniform spread of disease, the initial and the five interval plots in each replication were inserted with the susceptible RD6. Fertilizer was applied at a rate of 23 kg/ha (N: P: K) at 30 and 60 days after planting. Plants were protected through hand weeding and the use of common pesticides. Field water was maintained during the tillering stage at around 10 cm until ten days before harvesting. Daily minimum and maximum temperature, rainfall, and relative humidity during the experiment period, 2015–2018, were recorded.

### 2.2. Evaluation for Leaf Blast, Neck Blast, and Bacterial Blight Diseases Resistance

Leaf blast, neck blast, and BB disease reactions were observed individually from the inner ten plants, and scored through the Standard Evaluation System (IRRI, 1996) [16] on a scale of 0–9. Leaf blast

and bacterial blight were evaluated at the tillering stage, and neck blast disease evaluations were done at the ripening stage. Individual scores were averaged. Lines with a score of 0–3 were considered resistant, 4–5 moderately resistant, 6 moderately susceptible, and 7–9 susceptible.

**Table 1.** List of pyramided lines, and parent and check varieties used in this study.

| Line/Cultivar Name | Types of Lines | QTL on Chromosome [a] |
|---|---|---|
| BC$_2$F$_3$ 2-7-5-36 [c] | Breeding line | Blast (1, 2, 11, 12) and *xa5* |
| BC$_2$F$_3$ 2-8-2-36 [c] | Breeding line | Blast (1, 2, 11, 12) and *xa5* |
| BC$_2$F$_3$ 2-7-5-43 [c] | Breeding line | Blast (1, 2, 11, 12) and *xa5* |
| BC$_2$F$_3$ 2-8-2-19 [c] | Breeding line | Blast (1, 2, 11, 12) and *xa5* |
| BC$_2$F$_3$ 2-8-2-25 [c] | Breeding line | Blast (1, 2, 11, 12) and *xa5* |
| BC$_2$F$_3$ 9-1/15-1-28 [b] | Breeding line | Blast (1, 2,11,12) |
| BC$_2$F$_3$ 2-8-2-27 [c] | Breeding line | - |
| BC$_2$F$_3$ 2-8-2-52 [c] | Breeding line | Blast (1, 2, 11, 12) and *xa5* |
| BC$_2$F$_3$ 2-8-2-24 [c] | Breeding line | Blast (1, 2, 11, 12) and *xa5* |
| BC$_2$F$_3$ 6-1/15-2-11 [b] | Breeding line | Blast (1, 2, 12) |
| RD6 | Recurrent parent | - |
| Jao Hom Nin | Donor parent | Blast (1, 11) |
| P0489 | Donor parent | Blast (2, 12) |
| IR62266 | Donor parent | *xa5* |
| KDML105 | Susceptible check | - |
| PLT | Susceptible check | - |
| IR64 | Resistance check | - |
| PLD | Resistance check | - |

Note: [a] The number in round brackets indicates the location of the resistant QTL. [b] Suwannual et al. [14]; [c] Pinta et al. [15].

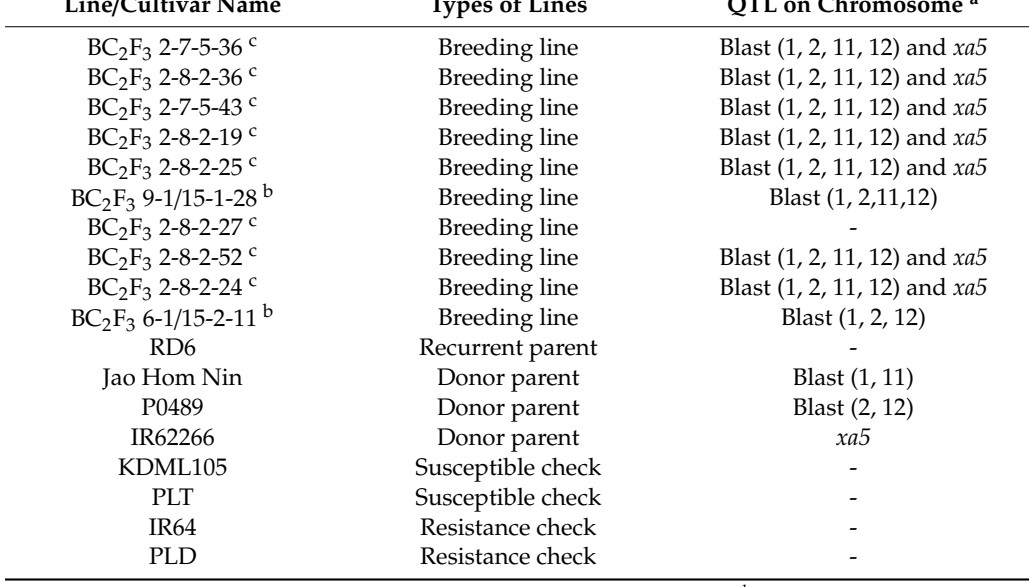

**Figure 1.** Schematic diagram of the farm phenotypic-based selection in multilocation target environments (2015–2018). AP, Ampawan farm; KKU, Khon Kaen University; KRC, Khon Kaen Rice Research center.

## 2.3. Evaluation for Agronomic Performance and Grain Yield

The agronomical characteristics were observed from the Khon Kaen University site, planted in 2018. Agronomical traits were measured according to the same evaluation system, in which four plants between the second and sixth row in each plot were selected for the following agronomic traits: days

to (50%) flowering (DTF), plant height (PH), number of tillers (NT), number of panicles (NP), panicle length (PL), panicle neck length (PNL), branching per panicle (BPP), filled grains per panicle (FGPP), unfilled grains percentage per panicle (UFG%), 1000-grain weight (GW), harvest index (HI), and grain yield (GY). All data were compiled through the use of raw measurements for data analysis except UFG% (UFG% = (number of unfilled grains per panicle/number of all grains per panicle) × 100) and HI (HI = total grain weight/total plant dry weight), which were calculate prior to data analysis.

### 2.4. Evaluation for Grain and Cooking Quality

Characterization of grain and cooking quality traits was performed on the grains harvested from the $BC_2F_5$ generation, Khon Kaen University site, planted in 2018. For grain characteristics, such as length, breadth, and length breadth ratio (L/B), data were obtained from ten fully-developed paddies and milled rice grain, and the means were again calculated. For cooking quality, the gelatinization temperature (GT) was measured indirectly by alkali spreading value (ASV) using six milled grains, as suggested by Little et al. [17]. Fragrance was determined following the sensory test protocol of Yi et al. [18], and the percentage of amylose content (AC %) was estimated following the method of Juliano [19]. The percentages of milled rice recovery (MRR %) and head rice recovery (HRR %) were calculated using 100g of paddy grains in each line, milled in a mini-milling machine, as per the formula given by SES, IRRI [16].

### 2.5. Sensory Analysis

Milled rice samples weighing 300g were washed three times and soaked for six hours. All samples were cooked in Thai Bamboo sticky rice steamers for 40 minutes, as is the local method. Textures were evaluated twice, i.e., in the morning (9:00 am) and in the afternoon (1:00 pm). Twelve local consumers, five males and seven females, were asked to measure the texture of the cooked rice, utilizing a three points hedonic scale: 1 = low, 2 = middle, and 3 = high. Their overall preferences were based upon their like or dislike of the rice, as well as their reasons. The mean scores from each individual consumer were compared.

### 2.6. Data Analysis

The traits examined in each experiment were subjected to statistical analysis via the Statistics $10^{\copyright}$ (1985–2013) program (Analytical Software, Tallahassee, FL, USA), which utilized replication as a random effect and genotypes as a fixed effect within the statistic model. Multiple comparisons were analyzed by Duncan's Multiple Range Test (DMRT) using the Statistical Tool for Agricultural Research (STAR) software (http://bbi.irri.org/products), developed by the IRRI's Biometrics and Breeding Informatics team.

## 3. Results

### 3.1. Evaluation of Leaf Blast, Neck Blast, and Bacterial Blight Resistance

Based on the combined ANOVA results, both leaf blast and neck blast diseases presented highly significant differences in infection levels among the different environmental conditions; however, the G × E interactions proved to be highly significant only for leaf blast disease (Table 2). Evaluations of the ILs, along with the recurrent parent (RP) RD6, donor parent (DP), and check lines for leaf blast, neck blast, and bacterial blight disease reaction during the 2015–2018 seasons, are shown in Tables 3 and 4. In 2015, both leaf blast and neck blast were infected at Ampawan (AP). The resulting blast scores showed that the selected ILs with resistant QTLs/genes presented complete resistance reactions for leaf blast (0.33–1.70) and neck blast (1.10–1.97), similar to those of both donor parents JHN (0.60) and P0489 (1.05), as well as the IR-64 (resistant check) (0.80), whereas the RP RD6 was displaced as susceptible (6.57), i.e., similar to that of the KDML105 (susceptible check), with a score of 6.43. However, in the

nondisease year of 2016 at AP, leaf blast infected all of the study genotypes, including RD6, in which the susceptible check lines produced a score of two or less.

**Table 2.** Summary of ANOVA results mean square values and percentage of the sum of squares (in parenthesis) for leaf blast, neck blast, and the grain yield variable for combined environment/location.

| SOV | df | Leaf Blast | Neck Blast | Grain Yield (kg ha$^{-1}$) |
|---|---|---|---|---|
| Environment (E) [1] | 4 | 27.5728 ** (11.21) | 31.4693 **(4.56) | 95094161.66 ** (57.56) |
| Replication/Envi. | 10 | 0.6354 (0.69) | 1.2839 (0.74) | 2261223.22 (3.42) |
| Genotype (G) | 10 | 23.4592 **(54.07) | 25.7281 **(63.41) | 662851.63 (1.00) |
| G × E | 39 | 4.1482 **(27.56) | 3.5731 (7.77) | 2093750.29 (12.36) |
| Error | 97 | 0.3647 **(6.48) | 2.5744 (23.51) | 1748007.61 (25.66) |

Note: The number above the traits is the number of the environment, ** Significant at 0.01 probability levels, respectively. [1] Not including the AP18 location, which flooded before harvesting.

We also found a serious BB infection at AP in 2017, in which the IL BC$_2$F$_3$ 2-8-2-52 produced the lowest BB infection score (3.33) compared to the other ILs and the susceptible lines, including RP RD6 (Table 4). Surprisingly, the DP IR62266 used for introgression *xa5* displayed a susceptible rating (6.00), similar to that of the RP RD6 (6.67) and susceptible check KDML105 (6.00). Furthermore, the JHN, used in introgression blast resistance genes, proved resistant to BB (2.00), with less infection than the multidisease resistant IR64 (3.33).

**Table 3.** Disease reaction scores for leaf blast in different years, under different environmental conditions, and at different locations.

| Genotype [t] | QTL on Chromosome | Leaf Blast | | | | Mean |
|---|---|---|---|---|---|---|
| | | AP15 | AP16 | AP18 | KKU18 | |
| BC$_2$F$_3$ 2-7-5-36 | Blast (1, 2, 11, 12) and *xa5* | 0.87$^{cd}$ | 0.03$^c$ | 0.00$^c$ | 0.07$^c$ | 0.24$^b$ |
| BC$_2$F$_3$ 2-8-2-36 | Blast (1,2, 11, 12) and *xa5* | 0.33$^d$ | 0.03$^c$ | 0.00$^c$ | 0.00$^c$ | 0.09$^b$ |
| BC$_2$F$_3$ 2-7-5-43 | Blast (1,2, 11, 12) and *xa5* | 0.40$^{cd}$ | 0.03$^c$ | 0.00$^c$ | 0.03$^c$ | 0.12$^b$ |
| BC$_2$F$_3$ 2-8-2-19 | Blast (1,2, 11, 12) and *xa5* | 0.83$^{cd}$ | 0.07$^c$ | 0.00$^c$ | 0.00$^c$ | 0.23$^b$ |
| BC$_2$F$_3$ 2-8-2-25 | Blast (1,2, 11, 12) and *xa5* | 0.77$^{cd}$ | 0.03$^c$ | 0.17$^c$ | 0.07$^c$ | 0.26$^b$ |
| BC$_2$F$_3$ 9-1/15-1-28 | Blast (1, 2, 11, 12) | 1.70$^c$ | 0.03$^c$ | 0.00$^c$ | 0.00$^c$ | 0.43$^b$ |
| BC$_2$F$_3$ 2-8-2-52 | Blast (1,2, 11, 12) and *xa5* | 1.03$^{cd}$ | 0.03$^c$ | 0.00$^c$ | 0.00$^c$ | 0.27$^b$ |
| BC$_2$F$_3$ 2-8-2-24 | Blast (1,2, 11, 12) and *xa5* | 0.87$^{cd}$ | 0.13$^c$ | 0.00$^c$ | 0.07$^c$ | 0.27$^b$ |
| BC$_2$F$_3$ 6-1/15-2-11 | Blast (1, 2, 12) | 0.47$^{cd}$ | 0.10$^c$ | 0.00$^c$ | 0.00$^c$ | 0.14$^b$ |
| BC$_2$F$_3$ 2-8-2-27 | No QTL | 5.10$^b$ | 1.03$^b$ | 2.33$^b$ | 4.07$^b$ | 3.13$^a$ |
| RD6 (RP) | - | 6.57$^a$ | 1.77$^a$ | 4.03$^a$ | 4.73$^b$ | 4.28$^a$ |
| Jao Hom Nin (DP) | Blast (1, 11) | 0.60$^{cd}$ | 0.03$^c$ | 0.00$^c$ | 0.00$^c$ | 0.16$^b$ |
| P0489 (DP) | Blast (2, 12) | 1.05$^{cd}$ | 0.10$^c$ | 0.00$^c$ | 0.00$^c$ | 0.29$^b$ |
| IR62266 (DP) | *xa5* | 0.47$^{cd}$ | 0.10$^c$ | 0.00$^c$ | 0.00$^c$ | 0.14$^b$ |
| PLD (LRC) | - | ND | 0.10$^c$ | 0.00$^c$ | 0.77$^c$ | 0.29$^b$ |
| IR64 (IRC) | - | 0.80$^{cd}$ | 0.07$^c$ | 0.00$^c$ | 0.00$^c$ | 0.22$^b$ |
| KDML105 (SC) | - | 6.43$^a$ | 1.10$^b$ | 0.53$^c$ | 7.53$^a$ | 3.89$^a$ |
| PLT (SC) | - | ND | 0.27$^c$ | 0.10$^c$ | 6.33$^a$ | 2.23$^a$ |
| Mean | | 1.77 | 0.28 | 0.4 | 1.32 | 0.959 |
| F-Test | | ** | ** | ** | ** | ** |
| CV (%) | | 38.0 | 104.5 | 147.5 | 57.9 | 119.6 |

Note: RP: recurrent parent; DP: donor parent; LRC: local resistance check; IRC: international resistance check; SC: susceptible check; ND: no data; KKU18: Khon Kaen University during 2018; AP15, 16, 18, Ampawan during 2015, 2016, and 2018; CV (%): coefficient of variation percentage; [t]: the generation of introgression lines were BC$_2$F$_3$, BC$_2$F$_4$, and BC$_2$F$_4$ in 2015, 2016, and 2018, respectively. ** Significant at $p \leq 0.05$. The letter after each value indicates the significance within each column by DMRT. Disease reaction scores were evaluated following the Standard Evaluation System [16].

**Table 4.** Disease reaction scores for neck blast and bacterial blight in different years, under different environmental conditions, and at different locations.

| Genotype [1] | QTL on Chromosome | Neck Blast | | Mean | BB |
| | | AP15 | KKU18 | | AP17 |
|---|---|---|---|---|---|
| $BC_2F_3$ 2-7-5-36 | Blast (1, 2, 11, 12) and *xa5* | 1.33$^c$ | 0.00$^c$ | 0.67$^d$ | 5.00$^{abc}$ |
| $BC_2F_3$ 2-8-2-36 | Blast (1, 2, 11, 12) and *xa5* | 1.97$^c$ | 0.00$^c$ | 0.99$^d$ | 6.00$^{ab}$ |
| $BC_2F_3$ 2-7-5-43 | Blast (1, 2, 11, 12) and *xa5* | 1.80$^c$ | 0.00$^c$ | 0.90$^d$ | 6.00$^{ab}$ |
| $BC_2F_3$ 2-8-2-19 | Blast (1, 2, 11, 12) and *xa5* | 1.70$^c$ | 0.00$^c$ | 0.85$^d$ | 5.33$^{abc}$ |
| $BC_2F_3$ 2-8-2-25 | Blast (1, 2, 11, 12) and *xa5* | 1.10$^c$ | 0.00$^c$ | 0.55$^d$ | 6.00$^{ab}$ |
| $BC_2F_3$ 9-1/15-1-28 | Blast (1, 2,11, 12) | 1.00$^c$ | 0.00$^c$ | 0.50$^d$ | 4.33$^{a-d}$ |
| $BC_2F_3$ 2-8-2-52 | Blast (1, 2, 11, 12) and *xa5* | 1.47$^c$ | 0.33$^c$ | 0.90$^d$ | 3.33$^{cd}$ |
| $BC_2F_3$ 2-8-2-24 | Blast (1, 2, 11, 12) and *xa5* | 1.70$^c$ | 0.00$^c$ | 0.85$^d$ | 6.00$^{ab}$ |
| $BC_2F_3$ 6-1/15-2-11 | Blast (1, 2, 12) | 1.30$^c$ | 0.00$^c$ | 0.65$^d$ | 5.33$^{abc}$ |
| $BC_2F_3$ 2-8-2-27 | No QTL | 5.43$^b$ | 5.07$^b$ | 5.25$^{bc}$ | 4.67$^{abc}$ |
| RD6 (RP) | - | 6.53$^b$ | 5.37$^b$ | 5.95$^b$ | 6.67$^a$ |
| Jao Hom Nin (DP) | Blast (1, 11) | 1.30$^c$ | 0.00$^c$ | 0.65$^d$ | 2.00$^d$ |
| P0489 (DP) | Blast (2, 12) | 1.96$^c$ | 0.00$^c$ | 0.98$^d$ | 4.33$^{cd}$ |
| IR62266 (DP) | *xa5* | 1.10$^c$ | 0.00$^c$ | 0.55$^d$ | 6.00$^{ab}$ |
| PLD (LRC) | - | ND | 0.00$^c$ | 0.00$^d$ | 4.00$^{bcd}$ |
| IR64 (IRC) | - | 1.30$^c$ | 0.00$^c$ | 0.65$^d$ | 3.33$^{a-d}$ |
| KDML105 (SC) | - | 8.23$^a$ | 8.27$^a$ | 8.25$^a$ | 6.00$^{ab}$ |
| PLT (SC) | - | ND | 5.77$^b$ | 5.77$^b$ | 3.67$^{bcd}$ |
| Mean | | 2.45 | 1.38 | 2.01 | 4.89 |
| F-Test | | ** | ** | ** | ** |
| CV (%) | | 28.5 | 151.57 | 18.9 | 25.54 |

Note: RP: recurrent parent; DP: donor parent; LRC: local resistance check; IRC: international resistance check; SC: susceptible check; ND: no data; ns: not significant; BB: Bacterial Blight; KKU18: Khon Kaen University during 2018; AP15, 17, Ampawan in 2015 and 2017; CV (%): coefficient of variation percentage; [1]: The generation of introgression lines were $BC_2F_3$, $BC_2F_4$, and $BC_2F_4$ in 2015, 2017, and 2018, respectively. ** Significant at $p \leq 0.05$. The letter after each value indicates the significance within each column by DMRT. Disease reaction scores were evaluated following the Standard Evaluation System [15].

In 2018, leaf blast and neck blast infected the site at KKU, whereas leaf blast alone occurred at AP (Tables 3 and 4). The infections which occurred at KKU were stronger than those at AP. At KKU, the ILs with resistant QTLs/genes were found to be completely resistant to both leaf blast (0.00–0.07) and neck blast (0.00–0.33), whereas the RP RD6 produced disease reaction scores of 4.73 for leaf blast, and 5.37 for neck blast. The RD6 produced a higher infection rate (4.03) than that of the KDML105 susceptible check line (0.53) and PLT (0.10) at AP; however, the ILs with resistant QTLs/genes were found once again to have complete resistance (0.00–0.17), similar to the donor and resistant check IR64. In general, the selected ILs with resistant QTLs/genes were found to be completely resistant for the leaf blast and neck blast, whereas only the IL $BC_2F_3$ 2-8-2-52 demonstrated resistance to both blast and BB diseases.

*3.2. Agronomic and Yield Performance of the Introgression Lines with RD6*

The agronomic performance of the ILs with RP RD6 is presented in Table 5. There were no significant differences ($p \leq 0.05$) between the selected ILs and RP RD6 for agronomic traits, TN, PN, PL, FGPP, TSW, and HI. The IL mean values ranged between 6.7–9.0 (TN), 5.7–7.3 (PN), 27.4–30.5cm (PL), 194.3–243.7 (FGPP), 26.6–28.7g (TSW), and 28.0–42.7% (HI). However, the ILs $BC_2F_3$ 2-8-2-19 and $BC_2F_3$ 2-7-5-36 showed early significance for (DTF) 103.0 and 116.0, respectively compared with the RD6 (136.7). Other ILs displaced ranges (131.0–135.0) similar to that of the RD6. Additionally, the PH of the ILs exhibited ranges (177.0–189.5) shorter than those of the RD6 (204.9). Moreover, all of the ILs were low for PNL (3.8–5.3) versus the RD6 (6.7). Unexpectedly, the UFG % of the ILs were higher (7.4–18.1) than the RD6 (6.6). In general, the selected ILs with resistant QTLs/genes, with the exception of the DTF, PH, PNL, and UFG % were similar to the original RD6.

The grain yield performances of the ILs with RP RD6 within the six environments are presented in Table 6. In 2016, the lower yield at AP may have been due to the high amount of rainfall during the tilling stage affecting the water level in the field as evidenced by the low tiller number. Additionally, the average GY at KKU in 2018 may have been lower due to the sandy soil conditions. Due to the late planting season, the rice yield was low again at the Rice Research Center (KRC) in 2018. The pooled mean of GY in the ILs with resistant QTLs/genes ranged from $BC_2F_3$ 2-8-2-19 (4566.66 kg/ha) to $BC_2F_3$ 2-8-2-25(5592.39 kg/ha), as opposed to the RD6 (4998.02 kg/ha). Some of the ILs exhibited production rates higher than the RD6, particularly the $BC_2F_3$ 2-8-2-25 and $BC_2F_3$ 2-7-5-36, which produced 5592.39 and 5400.37 kg/ha, respectively (Table 6). In general, most of the ILs have plant types similar to the original RD6, and are well adapted to the rain-fed lowland environment of northeast Thailand.

*3.3. Grain Quality, Cooking and Eating Quality of the Introgression Lines with RD6*

The superior grain quality of the ILs (Table 7) demonstrated grain characteristics similar to those of the RP RD6. The milled grain of the ILs with QTLs ranged 6.5–6.8 cm in length, and 2.1–2.2 cm in breath, with an L/B (3.00–3.1) similar to that of the RP RD6. Furthermore, the ILs with resistant QTLs/genes produced an MRR % (66.7–69.1) and HRR% (58.8–64.9) greater than that of the RD6 (67.1 and 58.9), respectively (Figure 2). The cooking quality results of the ILs with RP RD6 are presented in Table 7. The mean sensory fragrance of the ILs (1.5–2.3) was very near that of the RD6 (2.1). The ILs produced an ASV (4.9–5.7) or GT [(55–69) − (70–74)], similar to the RD6 ASV (5.3) or GT (70–74), as well as a low AC % (3.5–6.9), i.e., like that of the RP RD6 (5.2). This indicates that undesirable quality trait alleles were not located in QTLs 1, 2, 11, or 12, or in the *xa5*, which were introduced into the RD6 background. The eating quality sensory performances of the ILs with RD6 are compared in Table 8. Similarly, the morning and afternoon tests showed that IL $BC_2F_5$ 2-7-5-36 was preferred for its swelling capacity, sweetness, aroma, grain shape and size, and stickiness, being like that of the original RD6. The IL $BC_2F_5$ 2-8-2-52 scored second, followed by the other ILs (Table 8).

**Table 5.** Agronomic performance of introgression lines possessing blast- and bacterial blight-resistant genes/QTLs and the recurrent parent RD6.

| Genotype | Tiller | PN | PH (cm) | DTF | TSW (g) | PL (cm) | PNL (cm) | BPP | FGPP | UFG (%) | HI |
|---|---|---|---|---|---|---|---|---|---|---|---|
| $BC_2F_4$ 2-7-5-36 | 8.4 | 6.4 | 187.9[abc] | 116.0[bc] | 26.6 | 29.8 | 5.0[b] | 12.6 | 210.9 | 7.4[d] | 0.28 |
| $BC_2F_4$ 2-8-2-36 | 7.3 | 6.0 | 177.8[bc] | 131.0[ab] | 27.3 | 27.6 | 4.1[b] | 13.4 | 201.6 | 12.8[a-d] | 0.30 |
| $BC_2F_4$ 2-7-5-43 | 7.0 | 5.7 | 177.0[bc] | 132.7[ab] | 28.1 | 29.2 | 3.8[b] | 13.7 | 204.9 | 18.1[a] | 0.21 |
| $BC_2F_4$ 2-8-2-19 | 6.6 | 5.8 | 167.7[c] | 103.0[c] | 28.7 | 27.4 | 4.4[b] | 14.7 | 194.2 | 16.3[ab] | 0.43 |
| $BC_2F_4$ 2-8-2-25 | 7.0 | 6.6 | 189.5[ab] | 131.0[ab] | 27.3 | 30.5 | 4.6[b] | 14.6 | 243.4 | 11.3[a-d] | 0.29 |
| $BC_2F_4$ 9-1/15-1-28 | 8.0 | 6.1 | 187.4[abc] | 133.0[a] | 27.7 | 30.2 | 5.2[b] | 13.7 | 204.5 | 15.7[abc] | 0.36 |
| $BC_2F_4$ 2-8-2-27 | 7.6 | 6.0 | 204.6[a] | 135.0[a] | 27.4 | 28.8 | 5.3[b] | 13.7 | 198.2 | 9.6[bcd] | 0.29 |
| $BC_2F_4$ 2-8-2-52 | 9.0 | 7.1 | 184.1[bc] | 132.0[ab] | 27.7 | 29.5 | 4.0[b] | 13.9 | 197.6 | 8.1[cd] | 0.35 |
| $BC_2F_4$ 2-8-2-24 | 7.2 | 6.1 | 179.8[bc] | 130.7[ab] | 28.3 | 29.2 | 4.1[b] | 13.3 | 220.8 | 10.4[a-d] | 0.30 |
| $BC_2F_4$ 6-1/15-2-11 | 7.5 | 6.4 | 180.9[bc] | 132.7[ab] | 28.7 | 28.0 | 4.5[b] | 13.6 | 174.9 | 11.2[a-d] | 0.33 |
| RD6 (RP) | 9.0 | 6.3 | 204.9[a] | 136.7[a] | 27.4 | 29.0 | 6.7[a] | 12.5 | 201.3 | 6.6[d] | 0.25 |
| Mean | 7.7 | 6.2 | 185.6 | 128.5 | 27.7 | 29.0 | 4.7 | 13.6 | 204.8 | 11.59 | 0.31 |
| F-Test | ns | ns | * | ** | ns | ns | ** | ns | ns | * | ns |
| CV (%) | 12.34 | 8.46 | 5.78 | 6.71 | 3.14 | 4.62 | 16.06 | 6.16 | 12.7 | 34.49 | 28.7 |

Note: RP: recurrent parent; PN: panicle number per hill; PH (cm): plant height in centimeter; DTF: day to flowering; TSW (g): thousand seed weight in gram; PL (cm): panicle length in centimeter; PNL (cm): panicle neck length in centimeter; FGPP: number of filled grain per panicle; UFG (%): unfilled grain percentage per panicle; HI: harvest index; CV (%): coefficient of variation percentage; ns: * and ** are not significant and significantly different at $p \leq$ 0.05 and $p \leq$ 0.01, respectively. The letter after each value indicates the significance within each column by DMRT.

**Table 6.** Grain Yield (kg ha$^{-1}$) of introgression lines and the recurrent parent RD6 over different years and locations.

| Genotype [a] | Ampawan | | | 2018 | | Pool Mean |
|---|---|---|---|---|---|---|
| | **2015** | **2016** | **2018** | **KKU** | **KRC** | |
| BC$_2$F$_3$ 2-7-5-36 | 7573.47 | 3950.37 | 7730.00 | 2792.67 | 4955.33 | 5400.37 |
| BC$_2$F$_3$ 2-8-2-36 | 4787.87 | 3942.23 | 6370.00 | 3793.20 | 6070.00 | 4992.66 |
| BC$_2$F$_3$ 2-7-5-43 | 7953.33 | 2541.97 | 7003.33 | 3327.07 | 5599.47 | 5285.03 |
| BC$_2$F$_3$ 2-8-2-19 | 5510.40 | 3764.10 | ND | 3672.93 | 5319.20 | 4566.66 |
| BC$_2$F$_3$ 2-8-2-25 | 7718.53 | 2714.90 | 8440.00 | 3235.60 | 5852.93 | 5592.39 |
| BC$_2$F$_3$ 9-1/15-1-28 | 5444.93 | 3218.87 | 6770.00 | 4232.13 | 5525.07 | 5038.20 |
| BC$_2$F$_3$ 2-8-2-27 | 5574.53 | 4587.73 | 7473.33 | 3388.80 | 5389.87 | 5282.85 |
| BC$_2$F$_3$ 2-8-2-52 | 5252.00 | 2760.87 | 8903.33 | 4127.20 | 4606.00 | 5129.88 |
| BC$_2$F$_3$ 2-8-2-24 | 6746.13 | 3911.17 | 6653.33 | 3088.27 | 3862.53 | 4852.29 |
| BC$_2$F$_3$ 6-1/15-2-11 | 7019.87 | 3195.97 | 6386.67 | 3907.33 | 5305.60 | 5163.09 |
| RD6 (RP) | 6480.27 | 3799.03 | 7140.00 | 2953.87 | 4616.93 | 4998.02 |
| Mean | 6369.21 | 3489.75 | 7286.99 | 3501.73 | 5191.18 | 5118.31 |
| F-Test | ns | ns | ns | ns | ns | ns |
| CV (%) | 32.71 | 32.07 | 18.33 | 21.73 | 16.45 | 16.17 |

Note: [a] The generation of introgression lines were BC$_2$F$_3$, BC$_2$F$_4$, and BC$_2$F$_4$ in 2015, 2016, and 2018, respectively. RP, recurrent parent; KKU, Khon Kaen University; KRC, Khon Kaen Rice Research Center; ND, no data; ns, not significant; CV (%), coefficient of variation percentage.

**Table 7.** Cooking quality of introgression lines and the recurrent parent RD6.

| Genotype | QTL on Chromosome | Aroma | ASV | GT | AC |
|---|---|---|---|---|---|
| BC$_2$F$_5$ 2-7-5-36 | Blast (1, 2, 11, 12) and *xa5* | 2.3 | 5.7 | 55–69 | 3.5 |
| BC$_2$F$_5$ 2-8-2-36 | Blast (1, 2, 11, 12) and *xa5* | 1.8 | 5.7 | 55–69 | 5.4 |
| BC$_2$F$_5$ 2-7-5-43 | Blast (1, 2, 11, 12) and *xa5* | 1.8 | 4.9 | 70–74 | 6.9 |
| BC$_2$F$_5$ 2-8-2-19 | Blast (1, 2, 11, 12) and *xa5* | 1.7 | 5.3 | 70–74 | 4.7 |
| BC$_2$F$_5$ 2-8-2-25 | Blast (1, 2, 11, 12) and *xa5* | 2.0 | 5.3 | 70–74 | 5.1 |
| BC$_2$F$_5$ 9-1/15-1-28 | Blast (1, 2, 11, 12) | 1.9 | 5.3 | 70–74 | 4.7 |
| BC$_2$F$_5$ 2-8-2-27 | No QTL | 2.0 | 5.7 | 55–69 | 6.5 |
| BC$_2$F$_5$ 2-8-2-52 | Blast (1, 2, 11, 12) and *xa5* | 1.9 | 5.7 | 55–69 | 5.1 |
| BC$_2$F$_5$ 2-8-2-24 | Blast (1, 2, 11, 12) and *xa5* | 2.0 | 5.3 | 70–74 | 5.2 |
| BC$_2$F$_5$ 6-1/15-2-11 | Blast (1, 2, 12) | 1.5 | 5.3 | 70–74 | 5.3 |
| RD6 (RP) | - | 2.1 | 5.3 | 70–74 | 5.2 |
| Mean | | 1.9 | 5.4 | - | 5.2 |
| F-test | | ns | ns | - | ns |
| CV (%) | | 18.14 | 10.03 | - | 22.71 |

Note: RP: recurrent parent; ASV: alkali spreading value; GT: gelatinization temperature; AC: amylose content; ns: nonsignificant; CV (%): coefficient of variation percentage.

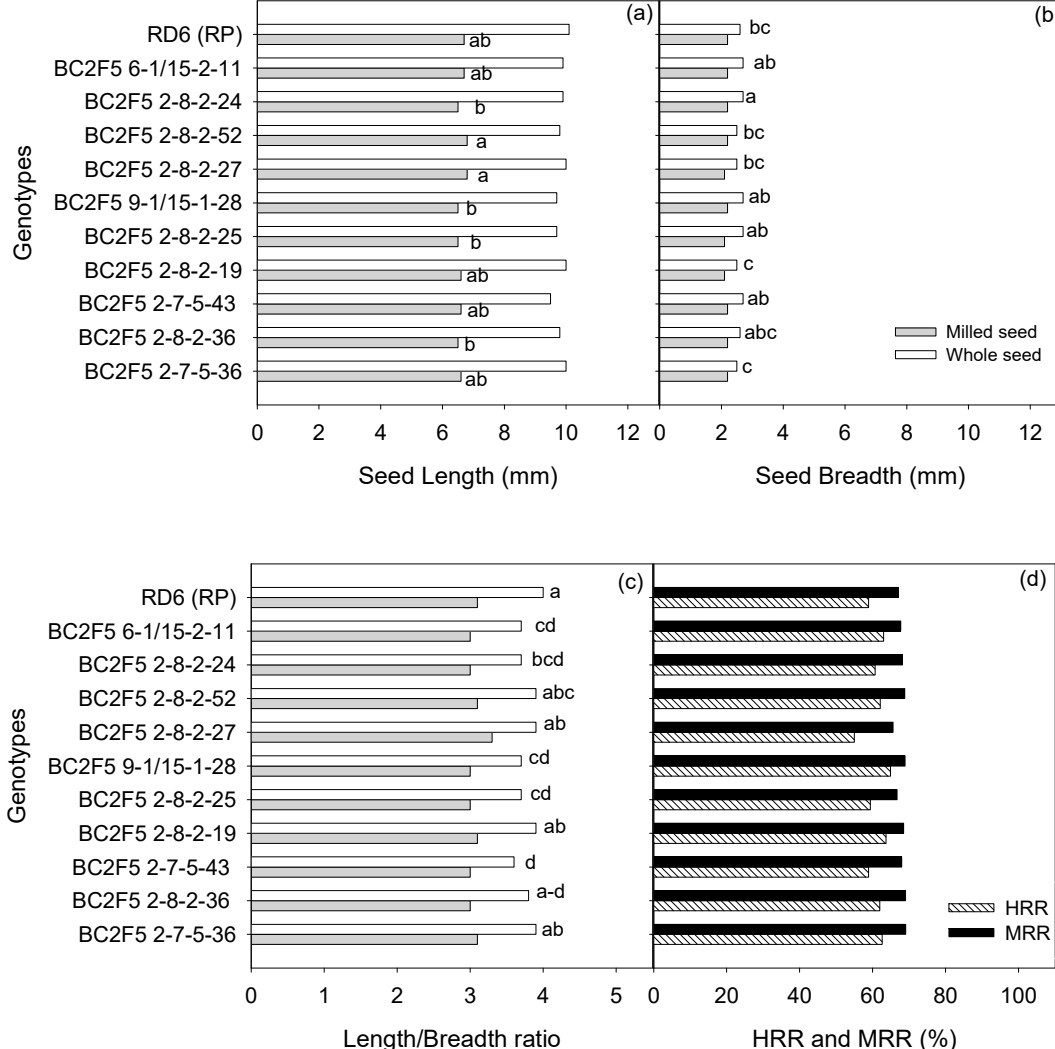

**Figure 2.** Grain quality attributes of introgression lines in comparison to the recurrent parent RD6; (**a**) seed length of milled and whole seed, (**b**) seed breadth of milled and whole seed, (**c**) seed length/breadth ratio of milled and whole seed, and (**d**) milled rice recovery percent (MRR) and head rice recovery percent (HRR); RP: recurrent parent; the different letter at each graph show significant different among genotypes in each trait at *p* < 0.05.

**Table 8.** Mean Sensory scores for four attribute eating qualities of the introgression lines with RD6.

| Genotype | Morning | | | | PR | Afternoon | | | | PR |
|---|---|---|---|---|---|---|---|---|---|---|
| | Aroma | Stiffness | Softness | Stickiness | | Aroma | Stiffness | Softness | Stickiness | |
| BC$_2$F$_5$ 2-7-5-36 | 1.9 | 1.8 | 1.9 | 2.1 | 1 | 1.7 | 1.0 | 1.6 | 1.8 | 1 |
| BC$_2$F$_5$ 2-8-2-36 | 2.0 | 1.5 | 1.8 | 2.2 | 3 | 1.6 | 1.7 | 1.2 | 1.5 | 5 |
| BC$_2$F$_5$ 2-7-5-43 | 1.7 | 1.5 | 1.4 | 1.3 | 4 | 1.8 | 1.6 | 1.5 | 1.5 | 3 |
| BC$_2$F$_5$ 2-8-2-19 | 1.8 | 2.3 | 1.9 | 1.8 | 4 | 1.6 | 2.2 | 1.6 | 1.6 | 6 |
| BC$_2$F$_5$ 2-8-2-25 | 1.6 | 1.7 | 1.3 | 2.0 | 5 | 1.6 | 1.8 | 1.0 | 1.0 | 7 |
| BC$_2$F$_5$ 9-1/15-1-28 | 1.6 | 1.5 | 1.8 | 1.7 | 4 | 1.7 | 1.7 | 1.5 | 1.0 | 5 |
| BC$_2$F$_5$ 2-8-2-27 | 2.1 | 1.0 | 2.5 | 2.3 | 2 | 1.8 | 1.0 | 2.0 | 1.6 | 2 |
| BC$_2$F$_5$ 2-8-2-52 | 1.5 | 1.0 | 2.4 | 1.9 | 2 | 1.7 | 1.0 | 1.9 | 1.7 | 2 |
| BC$_2$F$_5$ 2-8-2-24 | 1.5 | 2.0 | 1.8 | 1.9 | 3 | 1.6 | 1.5 | 1.6 | 1.8 | 5 |
| BC$_2$F$_5$ 6-1/15-2-11 | 1.6 | 1.4 | 1.6 | 1.8 | 4 | 1.5 | 1.4 | 1.5 | 1.8 | 4 |
| RD6 (RP) | 1.8 | 1.5 | 2.2 | 2.3 | 1 | 1.8 | 1.8 | 1.4 | 1.6 | 3 |

Note: RP: recurrent parent; PR: Preference rank, with 1 being the most appealing and 7 the least.

## 4. Discussion

Conventional breeding for disease resistance has proven to be time consuming and highly dependent on environmental conditions, in comparison to molecular breeding. The MAS, in particular, is simpler, more efficient, and accurate. For crop improvement in both durability and multiple disease resistance, gene pyramiding is emphasized to integrate many complex biochemical pathways within a plant [13]. However, the introgression of resistant genes without maintaining or improving yield levels and the grain quality would offer no reward; as such, varieties would not be adopted by farmers. Different MAS approaches are now, in practice, using background and foreground selection [20,21], as well as combined approaches utilizing foreground selection with stringent phenotypic selection [22], and foreground selection coupled with stringent phenotypic selection and background analysis [21]. Such practices have aided in accurate and efficient gene transfer, including gene pyramiding, into desired varietal backgrounds. In the present study, phenotypic selection was employed on the selected ILs for a higher degree of background recovery of the donor parent.

Blast disease caused by *M. oryzae* mainly occurs in two forms: seedling blast and panicle blast. In this study, the ILs with resistant QTLs genes showed outstanding resistance to leaf blast, as well as effective resistance against panicle neck blast (Tables 3 and 4). On the other hand, the ILs without resistant QTLs genes, the RP RD6, and the susceptible check lines, proved susceptible to both leaf blast and neck blast under favorable environmental conditions. The results obtained through multitrials indicated that lines lacking resistant QTLs genes were susceptible to both blast and BB diseases, as evidenced through blast conidia, brown spots, and yellow blight when grown in the open field under favorable weather conditions. Furthermore, the variations in disease incidence from one year to another indicated that weather conditions played an important role in both blast and BB incidence and development. Rainfall at the early flowering stages significantly affected blast disease, especially neck blast, which occurred only in 2015 and 2018. Although high amounts of rainfall occurred in the flowering stages in 2016, low plant density (low tiller number due to high water levers in the field) eradicated neck blast disease (Figure 3b). Rainfall and relative humidity are the most important factors affecting the sporulation, release, and germination of blast conidia as the microclimate within the plant canopy [10,23]. Interestingly, Jao Hom Nin, the donor parent for the blast resistance QTL on chromosome 1 and 11, also proved to be resistant to bacterial blight (Table 4). This may be because the gene contains a form of resistance to some BB pathogens strains, further indicating that this variety may be useful as a multidisease resistance source.

Incredibly, DP IR62266, used for introgression *xa5* gene, as well as most of the ILs studied, were susceptible to BB diseases, perhaps because a single resistance gene was introgressed for resistance against BB, and easily overcome by the pathogen, which can be explained by the *gene-for-gene* theory [24]. Furthermore, Kosawang et al. [25] reported that the pathogen is highly diverse in nature, particularly in Thailand, and several studies have indicated that cultivars with single BB-resistant genes do not provide broad-spectrum resistance [26–28]. Because several races of a pathogen within a rice-growing region can breakdown resistance, we have attempted to identify a new source of BB resistant genes (or combinations of *xa5* with other BB resistant genes) necessary for the further improvement of rice varieties in Thailand.

In the present study, we deployed a phenotypic selection for agromorphological traits in the later generations to identify backcross-resistant plants which are not only the closest to RD6, but also superior to the elite mega-varieties. Data from multilocations and numerous trial years demonstrated that the ILs had a potential yield similar or higher than the original RD6 (Table 6), as well as being well adapted to the rain-fed lowland environment in Thailand. We found that the ILs $BC_2F_3$ 2-8-2-25, $BC_2F_3$ 2-7-5-36, and $BC_2F_3$ 2-7-5-43 produced higher grain yields (5592.39, 5400.37, and 5285.03 kg/ha, respectively) than those of the RP RD6 and other ILs through each season. Also, the percentage of milling and head rice recovery were more displaced than the RP RD6 (Figure 2). Furthermore, despite an incidence of blast infection at both AP (2015) and KKU (2018), rice yields were still higher than those of the RP RD6 and the nonresistant ILs (Table 6). This may be due to the similar genome content of the

line with the recipient parent and the lower severity index of the blast and bacterial blight diseases of the genotype, as compared to the RP RD6. This also implies that the MAS is an effective RP RD6 approach to improve resistance to biotic stress in Jasmine rice [14,15]). These high levels of resistance to blast and BB, as well as the variations in yield potential, are a successful example of the integrated approach of selection at both the molecular and phenotypic level.

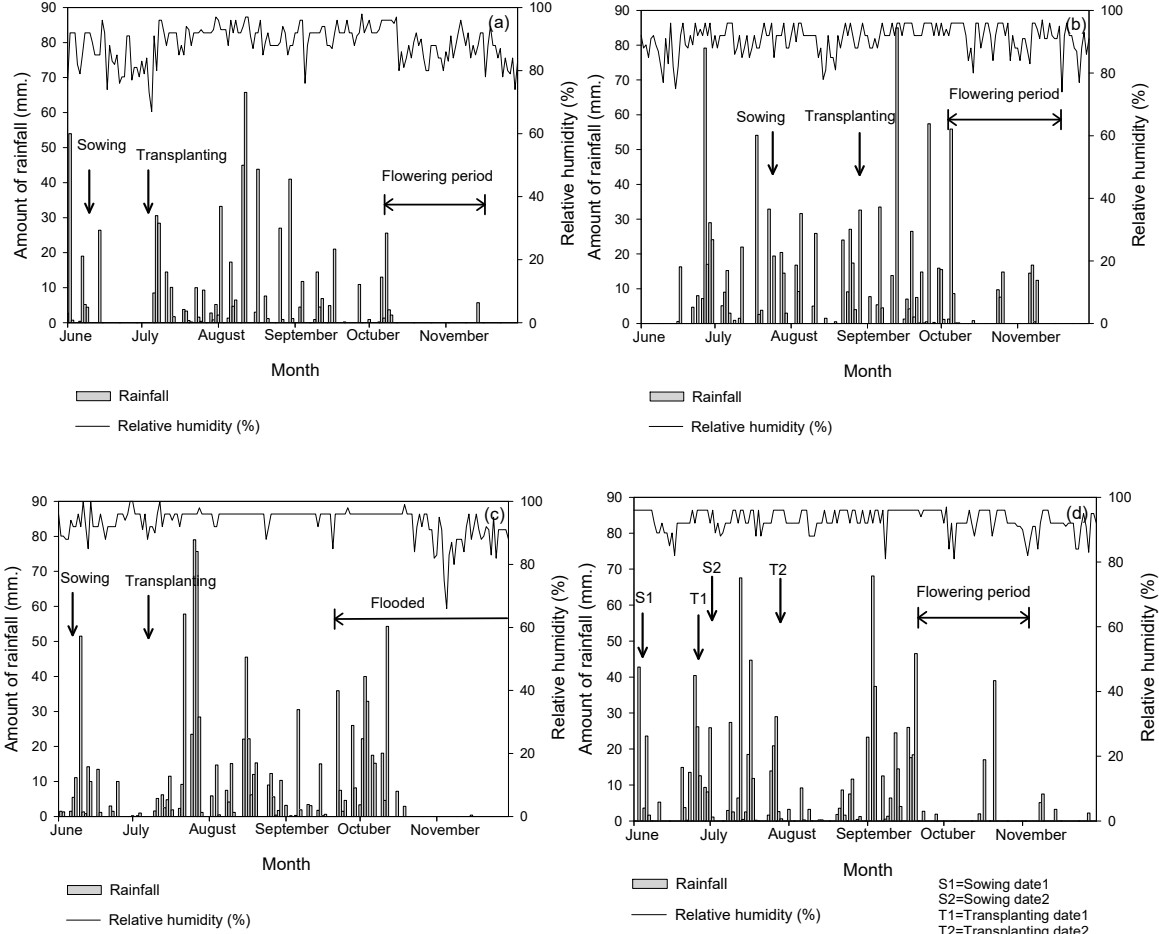

**Figure 3.** Daily minimum and maximum temperature, rainfall, and relative humidity during the experimental years of 2015(**a**), 2016(**b**), 2017(**c**), and 2018(**d**).

As expected, most of the agronomical and quality traits (TN, PN, TSW, FGPP, PL, BPP, HI, GL, GW, L/B, MRR %, and HRR %) of the ILs with resistant QTLs genes remained similar to those of the RD6 (Table 5 and Figure 2). However, some of the ILs showed slight differences to the RD6 for some traits, such as PH, DTF, PNL, and UFG% (Table 5). Panicle neck length, known as 'panicle exertion' (PE), is considered to be essential for enhancing total sink capacity and grain filling by improving the transport efficiency of assimilates [29]. Incomplete PE (also referred to as 'sheathed panicle') is an adverse symptom observed in rice plants which is associated with almost all cytoplasmic male sterile lines and which damages grain yield and raises disease incidence [30]. In our study, all of the ILs exhibited complete panicle neck lengths lower than those of the RD6 (Table 5) within similar parameters of ideal rice panicle types [31]. These agronomical trait variations might be due to a linkage drag of the donor segments with the target trait [32]. This is because some genes associated with undesirable traits are tightly linked with *R* genes on the rice genome [33].

Marker-assisted selection (MAS) is a highly-efficient breeding strategy for the introduction of disease *R* genes from both cultivated and exotic germplasms into adaptive genetic backgrounds across generations. However, if the donor parent used is a native landrace, the linkage drag problem may be

compounded [34]. In our study, three of the DPs are nonglutinous rice, possessing nonaromatic grains. However, we found that after the backcross of two terms, the most important factor determinants in grain and cooking quality, such as grain size and shape, aroma, GT, and AC (%), were comparable to those of the RP RD6 (Figure 2 and Table 7). They therefore achieved consumer preference due to their excellent swelling capacity, sweetness, aroma, grain shape and size, and stickiness, i.e., like that of the original RD6 (Table 8). Our research clearly proved that a cross between glutinous and nonglutinous varieties [14,15] was possible through the conversion of glutinous RD6 NIL, resulting in effective resistance to blast and BB, as well as possessing excellent qualities.

In general, it requires six to eight backcross cycles to gain the background of the recurrent parent. However, in later generations (i.e., after $BC_2$), it may be impossible to discriminate between backcross progeny and the RP based on individual plants [35]. In the present study, ILs $BC_2F_3$ 2-8-2-25, $BC_2F_3$ 2-7-5-43, $BC_2F_3$ 9-1/15-1-28, $BC_2F_3$ 2-7-5-36, and $BC_2F_3$ 2-8-2-52 expressed effective resistance to both leaf blast and neck blast. Some ILs showed resistant and moderately resistant reactions to BB disease, as well as having higher grain yields than those of the RP RD6. Furthermore, the agronomic characteristics and grain and cooking quality attributes were similar to those of the original RD6. Hence, we expect that it could be easily promoted among farmers who have shown a preference for the original RD6, thus negating the need to further backcross with RD6 again. Similar to the findings of Suwannual et al. [14] and Pinta et al. [15], we observed that the use of a pseudo-backcrossing design and MAB for two generations without genetic background selection was sufficient to recover the majority of the desired characteristics of the original RD6 within the $BC_2F_5$ generation.

The development of broad-spectrum and multiple disease resistance against blast and BB in Thailand is a major challenge, as rice cultivation areas have the presence of a number of genetically-distinct virulent pathogen strains in different geographical areas of the country. The present study demonstrated that the deployment of a four-QTL combination provided broad-spectrum resistance for blast, whereas a single *xa5* was unable to achieve durable and broad-spectrum resistance in many BB-prone rice growing areas in Thailand.

## 5. Conclusions

The ILs with resistant QTLs genes ($BC_2F_3$ 2-7-5-36, $BC_2F_3$ 2-7-5-43, $BC_2F_3$ 2-8-2-25, $BC_2F_3$ 2-8-2-52, and $BC_2F_3$ 6-1/15-2-11) are shown to be promising NILs, due to their superior characteristics in terms of yield, morphology, physico-chemical properties, and resistance. Our results indicate that $BC_2F_3$ 2-8-2-25 generated greater yield output compared to other ILs. It also produced long and slender grains, satisfactory milled and head rice recovery, aroma, GT, and an AC % similar to those of the original RD6. These lines also demonstrated better resistance against leaf blast and neck blast. However, only the IL $BC_2F_3$ 2-8-2-52 showed resistant against bacterial blight disease. Thus, these new rice lines deliver the prospect of successful combination with *xa5*, as well as other BB resistance genes, in order to achieve durable, broad-spectrum resistance in many BB-prone rice growing areas in Thailand.

**Author Contributions:** Data curation: M.S.A.N., J.J., A.S. and T.M.; Formal analysis: M.S.A.N. and S.C.; Methodology: M.S.A.N., T.M., S.C. and J.S.; Resources: S.C., T.M., and J.S.; Software: M.S.A.N. and S.C.; Supervision: S.C., T.M. and J.S.; Validation: M.S.A.N., S.C., T.M. and J.S.; Preparation of original draft: M.S.A.N. and S.C.; Writing, review, and editing: T.M., S.C. and J.S.

**Funding:** This research received no external funding.

**Acknowledgments:** This research was supported by The Plant Breeding Research Center for Sustainable Agriculture and The Salt-Tolerance Rice Research Group, Khon Kaen University, Khon Kaen, Thailand. Our gratitude is also extended to the Kachin State Agriculture Institute (Laiza) for providing financial support on behalf of Mr. Myo San Aung Nan.

**Conflicts of Interest:** The authors declare no conflict of interest.

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
