# Peer review of "Field Evaluation of RD6 Introgression Lines for Yield Performance, Blast, Bacterial Blight Resistance, and Cooking and Eating Qualities"

_agronomy, doi:10.3390/agronomy9120825_

Round 1

Reviewer 1 Report

This paper has done comprehensive field evaluation which is very good extension of lab bench work.

P2, L48; ‘Rice blast disease can infect all parts of the rice plant except the roots...’, but rice blast fungus can infect root as well according to the paper published in 2004 on Nature by Ane Sesma.

Figure2; Do figure 2 (a) and (e) corelated with the conclusions from this paper? It’s not necessary to have these pictures for the scientific paper.

Author Response

Dear reviewer,

Thank you very much for your comments and suggestions. We have revised our manuscript according to your comments and suggestions point by point. (Please see the revised manuscript and corrections file.)

Best regards

Sompong Chankaew

Reviewer 2 Report

The authors presented holistic results from a well-designed series of field experiments, demonstrating that they have identified introgression lines that have the potential to significantly increase disease resistance in an economically and socially important Thai rice cultivar. 

Lines 136-137: What was the exact model used? What variables were included as fixed or random effects? Were the traits normally distributed and used directly as the response in the model, or did the traits need to be transformed?

Figure 2 is unnecessary, as it does not show anything that has not been described in the text. I would recommend removing it.

Tables 2-3: Please describe the disease reaction scores presented here in more detail. Specifically, are these genotype means extracted from the statistical models, or are they grand means? What does "C. V. (%)" mean? I would also recommend including standard errors/deviations. 

Figure 3: This figure is not well described, but I am assuming that the purpose of this figure is to show differences in grain length and/or general size. If that is the case, this figure is unnecessary because Table 6 more fully describes differences in grain characteristics. If that is not the case and Figure 3 is describing information not presented in the other tables/figures, this figure needs to be described in greater detail.

Lines 285-287: This hypothesis needs to be tested with the data in the experiment. For example, rainfall and relative humidity could be included in the statistical models to assess whether they are significantly associated with disease incidence. As another example, statistical differences in rainfall and RH among years/locations could also be assessed.

Figure 4: This figure is unnecessary does not support the claim in lines 285-286. If this figure is presenting some other information not described in the text, it needs to be described in more detail.

Author Response

Dear reviewer,

Thank you very much for your comments and suggestions. We have revised our manuscript according to your comments and suggestions point by point. (Please see the revised manuscript and corrections file)

Best regards

Sompong Chankaew
